# Histologic Evaluation of the Effects of Folinic Acid Chitosan Hydrogel and Botulinum Toxin A on Wound Repair of Cleft Lip Surgery in Rats

**DOI:** 10.3390/jfb13030142

**Published:** 2022-09-05

**Authors:** Parastoo Namdar, Amirhossein Moaddabi, Rezvan Yazdian, Majid Saeedi, Fatemeh Ahmadian, Atena Shiva, Carmela Del Giudice, Parisa Soltani, Gianrico Spagnuolo

**Affiliations:** 1Department of Orthodontics, Faculty of Dentistry, Dental Research Center, Mazandaran University of Medical Sciences, Sari 48168, Iran; 2Department of Oral and Maxillofacial Surgery, Dental Research Center, Mazandaran University of Medical Sciences, Sari 48168, Iran; 3Faculty of Dentistry, Mazandaran University of Medical Sciences, Sari 48168, Iran; 4Department of Mollecular and Cell Biology Research Center, Faculty of Medicine, Mazandaran University of Medical Sciences, Sari 48168, Iran; 5Department of Pharmaceutics, Mazandaran University of Medical Sciences, Sari 48168, Iran; 6Student Research Committee, Faculty of Dentistry, Mazandaran University of Medical Sciences, Sari 48168, Iran; 7Department of Oral and Maxillofacial Pathology, Faculty of Dentistry, Mazandaran University of Medical Sciences, Sari 48168, Iran; 8Department of Neurosciences, Reproductive and Odontostomatological Sciences, University of Naples “Federico II”, 80131 Naples, Italy; 9Department of Oral and Maxillofacial Radiology, Dental Implants Research Center, Dental Research Institute, School of Dentistry, Isfahan University of Medical Sciences, Isfahan 81746, Iran

**Keywords:** cleft lip, folinic acid, botulinum toxin A, chitosan, wound healing

## Abstract

The aim of the present study was to compare the effects of folinic acid chitosan hydrogel and botulinum toxin A on the wound repair of cleft lip surgery in rat animal models. Cleft lip defects were simulated by triangular incisions in the upper lip of 40 Wistar rats. Then, the rats were randomly assigned to four groups: control (CTRL), chitosan hydrogel (CHIT), and folinic acid chitosan hydrogel (FOLCHIT), in which the wounds were covered by a gauze pad soaked in normal saline, chitosan hydrogel, and folinic acid chitosan hydrogel, respectively for 5 min immediately after closure; and botulinum toxin A (BOT) with the injection of 3 units of botulinum toxin A in the wound region. Fibroblast proliferation, collagen deposition, inflammatory cell infiltration, neovascularization, and epithelial proliferation and each parameter were rated on days 14 and 28. Statistical analysis was performed by Kolmogorov-Smirnov test, Shapiro-Wilk test, Kruskal-Wallis, and post-hoc tests (α = 0.05). The mean score for fibroblast proliferation was significantly higher in the FOLCHIT group compared with the BOT group at days 14 and 28 (*p* < 0.001, *p* = 0.012, respectively). At day 28, collagen deposition was significantly higher in the FOLCHIT group compared with the BOT group (*p* = 0.012). No significant difference was observed between the inflammatory infiltration of the study groups at the two time points (*p* = 0.096 and *p* = 1.000, respectively). At day 14, vascular proliferation of group FOLCHIT was significantly higher than groups CTRL and CHIT (*p* = 0.001 and *p* = 0.006, respectively). The epithelial proliferation in the FOLCHIT group was significantly higher than groups CHIT and CTRL at day 14 (*p* = 0.006 and *p* = 0.001, respectively) and day 28 (*p* = 0.012). In simulated lip cleft defects, topical application of folinic acid induces faster initial regeneration by higher inflammation and cellular proliferation, at the expense of a higher tendency for scar formation by slightly higher fibroblast proliferation and collagen deposition. While injection of botulinum toxin A provides less fibroblast proliferation and collagen deposition, and thus lower potential for scar formation compared with the folinic acid group. Therefore, in wounds of the esthetic zone, such as cleft lip defects, the application of botulinum toxin A shows promising results.

## 1. Introduction

Clefts of lip and palate are the most common congenital abnormalities of the craniofacial region with an average prevalence of one in every 700 newborns [1,2,3,4]. Surgical reparation techniques inevitably lead to scar formation as a result of pathologic remodeling [5]. Reducing the postsurgical scar and reaching a better esthetic outcome are the main goals in cleft reconstruction procedures. Precise incision design, anatomical reconstructions of musculature, and atraumatic suturing are crucial for obtaining minimum scar tissue. However, due to the constant construction of muscles and skin tension caused by the infant’s feeding and crying, the achievement of ideal esthetic outcomes after lip reconstruction is difficult [6].

Different pharmacological interventions can be used to prevent scar formation and help the wound repair process [7]. These treatments include injection of corticosteroid in the lesion, application of local retinoids, injection of interferons or chemotherapeutic agents in the lesion, and injection of bleomycin [8]. In recent years, injection of botulinum toxin A has been shown to be effective for the prevention and treatment of hypertrophic scar lesions [1,5,9]. Additionally, folinic acid, a derivative of folic acid naturally found in food, has been a subject of research for its wound healing effects [10,11].

Chitosan hydrogel scaffold is used as a synthetic cellular matrix for providing support for cellular reconstruction and repair [12]. Additionally, hydrogels possess the potential to carry small proteins, growth factors, and other agents crucial for growth, differentiation, and reparation. Using hydrogel scaffolds enhances tissue reparation by increasing the local concentration of growth agents in the wound site [13]. In oral surgery, chitosan-based hydrogel scaffolds have been used for regenerative purposes showing promising outcomes [14,15]. However, to the authors’ knowledge, no previous study has been performed on chitosan-based hydrogel scaffolds containing folinic acid in reparation of cleft surgery scars. Additionally, no study has compared the effects of folinic acid and botulinum toxin A in vivo in wound healing in cleft surgery. Therefore, the aim of the present study was to compare the effects of folinic acid chitosan hydrogel and botulinum toxin A on wound repair of cleft lip surgery in rat animal models.

## 2. Materials and Methods

The protocols of the present animal study were approved by the Research Ethics Committee of Mazandaran University of Medical Sciences (#IR.MAZUMS.REC.1399.785). A sample size of least 6 cases in each group was calculated while taking into account α = 0.05, β = 0.20, and a power test of 80%. A total of 40 Wistar rats weighing between 350 and 400 g, that were obtained from the research center of Mazandaran University of Medical Sciences, were used in this study.

### 2.1. Preparation and Characterization of Folinic Acid Chitosan Hydrogel

Chitosan hydrogel was prepared by dissolving 4 g of medium molecular weight of chitosan powder in 100 mL of 0.1% acetic acid (*v*/*v*, Merck, Darmstadt, Germany) The mixture was stirred for 1 h and sonicated for 10 min for the elimination of air bubbles. To load folinic acid into the hydrogel, 50 mg of folinic acid powder (Sigma-Aldrich, St. Louis, MO, USA) was dissolved in 1% acetic acid and stirred for 1 h. Additionally, 1 mL of prepared Br-AG-CS NPs solution containing 50 mg bromelain was centrifuged at 15,000 rpm for 20 min at room temperature. Then, the obtained NPs, re-suspended in 20 µL of deionized water were added to 1 mL of 3% chitosan hydrogel and stirred for 2 h.

The Fourier transform infrared spectroscopy (FTIR) study was carried out at a wavelength range of 4000-450 cm^−1^ using a spectrometer (PerkinElmer, Waltham, MA, USA). The thermal behavior of hydrogel was characterized by differential scanning calorimetry (DSC, Mettler Toledo, Columbus, OH, USA) at the temperature range between 20 and 350 °C at a heating rate of 10 °C min^−1^.

The determination of viscosity of 3% (*w*/*v*) chitosan hydrogel containing folinic acid was measured using a Brookfield R/S+ Rheometer (Brookfield, Middlebro, MA, USA) with SC4-25 spindle at a shear rate of 0–200 s^−1^.

Furthermore, in vitro release profile of folinic acid loaded into hydrogel was carried out in PBS, pH 7.4, then 1 mL of chitosan hydrogel containing 2.5% folinic acid (based on previous studies [11]) was transferred into the dialysis membrane and incubated in 3 mL of PBS solution at 37 °C. At different time points, aliquots were withdrawn and replaced with the same volume. The absorbance of these aliquots was measured at 270 nm using a UV-Vis spectrophotometer (Jasco V-630, Tokyo, Japan). The folinic acid concentration was measured using a calibration curve.

### 2.2. Preparation of Animal Models and Interventions

The rats used in this study were housed in a controlled room at an approximate temperature of 22 °C with free access to food and water.

Initially, the rats were anesthetized with an intraperitoneal injection of ketamine hydrochloride (0.5 mL) and xylazine hydrochloride (0.1 mL) for creating cleft lip defects. Then, the hair on the faces of rats were shaved, and triangular 7 mm × 7 mm × 4 mm incisions were created at the upper lip of rats on the left side. Thereafter, the wounds were sutured in three layers. The rats were randomly divided into four groups: The control group (CTRL), in which the wounds were covered by a normal saline-impregnated gauze pad for 5 min immediately after closure; intervention group 1 (BOT), in which 3 units of botulinum toxin A was injected in the wound region; in intervention group 2 (CHIT), the wounds were covered with a gauze pad soaked in chitosan hydrogel for 5 min immediately after closure; the wounds in intervention group 3 (FOLCHIT) were covered with a gauze pad soaked in folinic acid chitosan hydrogel for 5 min immediately after closure.

In each group, half of the rats were euthanized at day 14 and the other half were euthanized at day 28 after the last intervention.

### 2.3. Histopathologic Examination

For microscopic evaluation, cleft and surrounding tissue samples were obtained from the animals. To investigate the proliferation of fibroblast, collagen deposition, inflammation cells, proliferation of blood vessels, and proliferation of epithelial cells in the study groups, the wounds were collected, isolated, and placed in 10% formalin for 24 h. After tissue processing (fixating, dehydrating, paraffin embedding, and cutting), the slices were obtained with a thickness of 5 micrometers using a microtome (Leitz GmbH, Oberkochen, Germany). Hematoxylin and eosin staining was used to determine connective properties and Masson’s trichrome staining was applied to determine the amount of deposited collagen. For each animal model, one slide with each of these stains was prepared. Finally, fibroblast proliferation, collagen deposition, inflammatory cell infiltration, neovascularization, and epithelial proliferation and each parameter were assessed using a semiquantitative approach and rated using the scale 0: None, 1: Mild, 2: Moderate, 3: Marked on days 14 and 28. Moreover, the presence or absence of necrosis and abscess in the tissues was examined in this study.

To investigate the degree of inflammation, the inflammatory cells, including neutrophils, macrophages, plasma cells, and lymphocytes were counted three times, at 400× magnification with a microscope (Nikon-Eclipse-E100, Minato, Japan). Moreover, the number of fibroblasts and fibrocytes was counted to analyze the rate of granulation tissue formation and maturation of the healing area. Vascular proliferation was measured at 400× to evaluate the extent of angiogenesis in the study groups. In Masson’s trichrome staining, the amount of mature collagen with a deep blue color and new collagen with a light blue color was analyzed using the same scoring of hematoxylin and eosin staining.

All of the examinations were performed by an experienced oral pathologist (with 13 years of experience) who was blinded to the clinical topographic aspects of the wounds as well as the grouping.

### 2.4. Statistical Analysis

The collected data were analyzed in Statistical Package for the Social Sciences (SPSS, version 25, IBM Statistics, Armonk, NY, USA) through descriptive statistics (frequency and mean ± SD). The Chi-square test was used to compare the nominal data. The normal distribution of data was examined using the Kolmogorov-Smirnov and Shapiro-Wilk tests. Since the data distribution was abnormal, the Kruskal-Wallis test was employed for ranking the qualitative data to make comparisons among the three groups. Moreover, post-hoc tests were utilized in this study. In all calculations, a *p*-value less than 0.05 was considered statistically significant.

## 3. Results

The FTIR, DSC, viscosity, and release profile graphs of the prepared folinic acid chitosan hydrogel are provided in Figure 1a–d, respectively. The intense peak in the FTIR diagram at 1553 confirms the presence of amide in the chemical structure of chitosan and the absence of some peaks of the folinic acid is due to physical entanglement with the hydrogel system (Figure 1a). The DSC thermograms of lyophilized folinic acid chitosan hydrogel showed the endothermic peak around 240 °C, while the endothermic peak of folinic acid was not observed in drug-hydrogel, demonstrating that the physical form of folinic acid was altered from crystalline to amorphous (Figure 1b). According to the viscosity profile of chitosan hydrogel and folinic acid chitosan hydrogel, the presence of folinic acid decreased viscosity in comparison with the plain hydrogel (Figure 1c). Physical interactions between the drug molecules and functional groups of chitosan decreased the viscosity. The FTIR data showed no chemical interactions between chitosan and folinic acid. These results are indicative of a slight decrease in hydrogel viscosity in the presence of folinic acid. The hydrogel preparation had a considerably low release profile. A burst release was observed during 4 h, showing that the hydrogel could modify the rate of drug release and provide a long-acting drug delivery mechanism (Figure 1d).

During the follow-up period, no evidence of abscess formation or necrosis was observed in the animal models. Figure 2 and Figure 3 show the samples of histopathologic hematoxylin and eosin images in each experimental group at days 14 and 28, respectively. Figure 4 demonstrates the histopathologic Masson’s trichrome image of a sample in the FOLCHIT group after 28 days.

### 3.1. Fibroblast Proliferation

After 14 days, the maximum and minimum mean scores for fibroblast proliferation were observed in groups FOLCHIT (3) and BOT (1.4), respectively. A significant difference was observed between the four groups (*p* < 0.001, Table 1). Pairwise analysis showed that the mean score for fibroblast proliferation is significantly higher in the FOLCHIT group compared with the BOT group (*p* < 0.001). However, no significant difference was observed in the pairwise analysis of the other groups (*p* > 0.05).

At day 28, the highest mean score for fibroblast proliferation belonged to the FOLCHIT group (2.4), while the lowest mean score belonged to the BOT group (1.4). A significant difference was observed between the four groups (*p* = 0.030, Table 1). However, based on the pairwise analysis, the only significant difference was observed between groups FOLCHIT and BOT (*p* = 0.012).

In comparison with the fibroblast proliferation in each group between days 14 and 28, groups CTRL, BOT, and CHIT had similar scores at both times. In group FOLCHIT, all samples had marked proliferation at day 14, while at day 28, fibroblast proliferation was marked in only 40% of the samples. However, this difference was not statistically significant (Table 1, *p* = 0.151).

### 3.2. Collagen Deposition

After 14 days, the highest and lowest mean scores for collagen deposition were observed in groups FOLCHIT (2.2) and BOT (1.6), respectively. No significant difference was observed among the study groups (Table 2, *p* = 0.103).

After 28 days, the maximum mean score for collagen deposition belonged to the FOLCHIT group (2.4), while the minimum score was observed in the BOT group (1.4). The difference among the study group was significant in collagen deposition at this time point (Table 2, *p* = 0.020). Pairwise analysis revealed a significant difference between groups FOLCHIT and BOT (*p* = 0.012). The difference in collagen deposition was not significantly different in the other groups (*p* > 0.05).

In comparison with the collagen deposition in each group between the two time points, no significant difference was observed (*p* > 0.05).

### 3.3. Inflammatory Cells Infiltration

At day 14, the mean score for infiltration of inflammatory cells in the FOLCHIT group was 2.4, while in all the other groups, the score was 2. The difference between these values was not statistically significant (Table 3, *p* = 0.096).

After 28 days, in all groups, the mean value for inflammatory cell infiltration was 1, indicating mild infiltration of inflammatory cells for all samples (Table 3, *p* = 1.000).

In comparison with the infiltration of inflammatory cells at days 14 and 28, a significant decrease was observed in the mean scores of all groups (Table 3, *p* = 0.008).

### 3.4. Vascular Proliferation

At day 14, regarding vascular proliferation, the highest and lowest mean scores belonged to groups FOLCHIT (3) and CTRL (1), respectively. The difference between the study groups was statistically significant (Table 4, *p* = 0.001). Pairwise comparison revealed statistically significant differences between group FOLCHIT and groups CTRL and CHIT (*p* = 0.001 and *p* = 0.006, respectively).

After 28 days, the maximum mean score for vascular proliferation was 1 in groups BOT and FOLCHIT and the minimum score was 0.6 in the CTRL group, showing no statistically significant difference among the four groups (Table 4, *p* = 0.251).

In comparison with the mean scores of vascular proliferation between the two time points, a significant reduction was observed in groups FOLCHIT and BOT (*p* = 0.008). The other groups did not show a statistically significant difference between days 14 and 28 (Table 4, *p* > 0.05).

### 3.5. Epithelial Proliferation

After 14 days, the maximum and minimum mean scores for epithelial proliferation were observed in groups FOLCHIT (3) and CTRL (1), respectively. A significant difference was observed among the four groups in epithelial proliferation (Table 5, *p* = 0.001). According to the pairwise analysis, the only significant difference was observed between the FOLCHIT group and groups CHIT (*p* = 0.006) and CTRL (*p* = 0.001).

At day 28, the mean score for epithelial proliferation was 3 in groups BOT and FOLCHIT and 2 in groups CTRL and CHIT (Table 5). The difference between the groups BOT and FOLCHIT and groups CTRL and CHIT was statistically significant (*p* = 0.012).

In comparison the mean values at days 14 and 28, a significant increase in epithelial proliferation scores was observed in groups CTRL, BOT, and CHIT (Table 5, *p* = 0.008, *p* = 0.008, and *p* = 0.032, respectively).

## 4. Discussion

Based on our findings, in terms of fibroblast proliferation and collagen deposition, the FOLCHIT group had the highest rank, while the BOT group showed the least mean score in fibroblast proliferation and collagen deposition at both time points. Additionally, the FOLCHIT group had the highest inflammatory infiltration at day 14, while at day 28, all groups showed a decrease in inflammation and showed similar counts of inflammatory cells. In vascular proliferation at day 14, the FOLCHIT group had the highest rank among the study groups. Although at day 28, the FOLCHIT and BOT groups had the highest vascular proliferation scores among the groups, the decrease in the scores of vascular proliferation between the time points was significant for these two groups. Finally, epithelial proliferation scores were higher in the FOLCHIT group and FOLCHIT and BOT groups at days 14 and 28, respectively. A significant increase in epithelial proliferation was observed in groups CTRL, BOT, and CHIT, while the FOLCHIT group had the maximum score at both time points. To the authors’ knowledge, no previous study has been performed in comparison with folinic acid and botulinum toxin A in wound healing. Therefore, a comparison of the results of the present study with other findings is not possible at this point.

Reconstruction of the upper lip and restoration of the continuity of the orbicularis oris muscle is an important part of treatment for individuals with cleft lip. However, scar formation after surgical procedures is inevitable. Improvement in the esthetic outcome of surgical reparative procedures by reducing the amount of scar is one of the major goals of corrective surgery of the upper lip. Different agents have been used in research settings for enhancing the wound healing process and reducing hypertrophic scar formation. In recent years, injection of botulinum toxin A has been experimented with positive findings for the prevention and treatment of hypertrophic scar lesions [1,5,9]. Additionally, folinic acid has been a subject of research for its wound healing effects [10,11].

Fibroblasts are responsible for the production of collagen, which in turn, is crucial for wound repair. At both time points, although not significantly different from the CTRL group, the FOLCHIT group had the highest amount of proliferation of fibroblasts and collagen deposition. Folic acid has been shown to improve fibroblast proliferation and collagen neosynthesis [16]. It can promote DNA repair in the fibroblasts of the dermis [17]. Based on the findings of an in vivo study, folic acid and creatinine increase collagen gene expression, procollagen levels, and collagen fiber density in fibroblasts [18]. In a study performed by Duman et al., the topical application of 2.5% folinic acid in rat animal models has caused an increase in fibroblastic proliferation and collagen deposition leading to accelerated wound healing [11]. Additionally, a synthetic folic acid gel showed enhanced wound repair in oral ulcers of rats and rabbits [19]. Nonetheless, excessive fibroblast proliferation and the resultant collagen accumulation lead to the formation of hypertrophic scar. Therefore, the higher fibroblast proliferation and collagen deposition are indicative of higher scar formation [20]. The BOT group showed the lowest mean score for collagen accumulation and fibroblast proliferation. The findings of Lee et al. showed that in week 4, the number of fibroblasts in skin wounds with an injection of botulinum toxin A was significantly lower compared with the control group [21]. In the present study, although no significant difference was found in the pairwise analysis of control and botulinum toxin A groups, the wounds with a Botox injection showed lower fibroblast proliferation scores at both time points. In contrast, the findings of Hussein et al. indicated that the injection of botulinum toxin A caused an increase in burn wound samples of rat models [22]. The difference between the findings can be due to the different natures and locations of the wounds. Although ultimately, Hussein et al. reported an improved cosmetic appearance in the simulated burn scars using botulinum toxin A.

Inflammation is another important part of the initiation of wound healing. The inflammatory phase starts immediately after the injury, lasting for about 3 days. Infiltration of inflammatory cells is responsible for prevention from microbial colonization and inflammatory cells remain in the site of the injury for weeks. Based on our findings, the highest rate of inflammatory infiltration belonged to the FOLCHIT group at day 14. The study by Duman et al. revealed the pro-inflammatory response of folinic acid, while the topical application of 2.5% folinic acid resulted in higher inflammatory infiltration in wounds compared with 1% folinic acid and dexpanthenol [11]. Based on our findings, the BOT group was not significantly different in terms of inflammatory infiltration compared with the control group. In the study by Lee et al., the degree of inflammation in the surgical wounds of rats treated with Botox was significantly less than the control group. In all of the study groups, mild infiltration of inflammatory cells was observed at day 28. During the final stages of wound healing, the number of inflammatory cells decreased and the local inflammation is resolved [23]. Moreover, our findings showed a significant decrease in the number of inflammatory cells at day 28 compared with day 14 in all study groups.

The process of vascular proliferation is stimulated by local factors, such as tissue hypoxia, low pH, and high levels of lactate [24]. Additionally, a variety of cells present at the injury site produce mediators that are potent angiogenic signals for endothelial cells. These factors cause migration of the endothelial cells of the capillaries adjacent to the wound site to the devascularized region. Subsequently, the endothelial cells proliferate and form vascular buds which evolve into a new vascular loop. This process continues until the newly formed local vascular system is sufficiently developed to meet the oxygenation and metabolic needs of the repairing tissue [25]. The findings of Duman et al. showed that folinic acid enhances neo-vessel formation in the rat wound models until day 14 [11]. This is consistent with our findings, as the FOLCHIT group had the highest rank in vascular proliferation compared with the other groups. However, at day 28, the FOLCHIT group showed the most significant reduction in angiogenesis. This can be justified by the anti-angiogenic effects of folic acid, which have been shown in the study by Lin et al. on cultured human endothelial cells [26]. However, more studies are needed to further clarify the effects of folinic acid on vascular proliferation during wound healing. The BOT group also showed a significant reduction in vascular proliferation at day 28 compared with day 14. Botulinum toxin A has been reported to improve angiogenesis in adipose tissue grafts [27]. Additionally, in a study performed by Kim et al. in rat cutaneous flaps, injection of botulinum toxin A caused a significant increase in endothelial proliferation and expression of angiogenesis growth factors, including vascular endothelial growth factor [28]. In another study on cells present in mature scars, Botox incubation did not affect cell proliferation of endothelial cells [29]. These findings are in contrast with our findings, reporting that the BOT group was not significantly different from the CTRL group in terms of vascular proliferation. This difference can be justified by different cell populations as well as different concentrations of botulinum toxin A. However, similar to our study, Lee et al. reported that vessel proliferation was not significantly different between the control wounds with the injection of normal saline and wounds with the injection of botulinum toxin A in all time points through 8 weeks in rats [21]. Further studies analyzing the effects of Botox with different concentrations and amounts can be considered.

Another aspect studied in the present study was epithelial proliferation. Epithelial proliferation is a multi-step process involving detachment and internal modification of cells, migration, proliferation, and differentiation. Mediators, such as epidermal growth factor, keratinocyte growth factor, and transforming growth factor, are responsible for the proliferation of basal epithelial cells during the healing process. In terms of epithelial proliferation, a significant difference was observed among the FOLCHIT group with CHIT and CTRL groups at day 14 and FOLCHIT and BOT groups with CHIT and CTRL groups at day 28. Epithelial proliferation increased significantly in all groups except for the FOLCHIT group, in which the highest score was observed at both time points. In a study by Xiao et al., incorporation of folic acid in a nanoparticle metal framework resulted in re-epithelialization in vivo [10]. Epithelialization is a clinical sign for successful healing, but it is not the final event. Remodeling of the granulation tissue is the ultimate step in the healing process [25].

Clinical application of Botox injection in the peri-oral region can lead to oral incompetence, speech difficulties, drooling, asymmetric smile and lip movements, and lip flattening. However, several authors support the injection of botulinum toxin A in the cleft region despite the potential complications. According to the findings of the study, it seems that folinic acid allows for a faster initial regeneration by higher inflammation and cellular proliferation, at the expense of a higher tendency for scar formation by slightly higher fibroblast proliferation and collagen deposition. Further studies can investigate the effects of the combined application of folinic acid and botulinum toxin A in cleft defects in animal models by incorporating the reported scar preventing properties of botulinum toxin A into the regenerative features of folinic acid.

The drug release pattern in hydrogel is an important parameter in evaluating its effectiveness. The profile in which the drug is released by the hydrogel is frequently critical to reach a desired treatment output, and suitable drug distribution (short-term vs. long-term) and its release characteristics (continuous vs. pulsatile) are dependent on the specialized usage [30]. The results of the release profile show a burst release during 4 h, which is indicative of the long-acting drug delivery mechanism of the prepared folinic acid chitsan hydrogel.

The present study had some limitations. Due to limited number of rats, only one concentration of folinic acid was incorporated into the chitosan hydrogel. Additionally, the response of rat animal models might not necessarily be the same as the response of human wound cells to the different experiments performed in this study.

## 5. Conclusions

In simulated lip cleft defects, the topical application of folinic acid induces faster initial regeneration by higher inflammation and cellular proliferation, at the expense of a higher tendency for scar formation by slightly higher fibroblast proliferation and collagen deposition. Meanwhile, the injection of botulinum toxin A provides less fibroblast proliferation and collagen deposition, and thus a lower potential for scar formation compared with the folinic acid group.

## Figures and Tables

**Figure 1 jfb-13-00142-f001:**
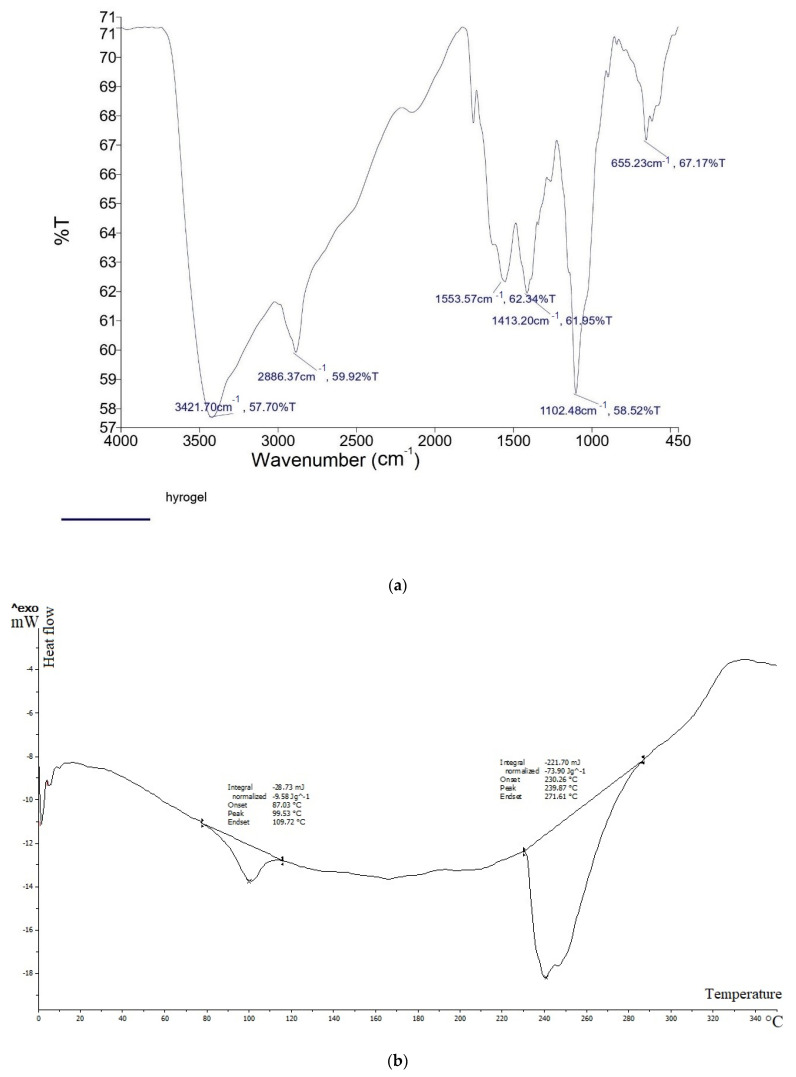
Diagrams of FTIR (**a**), DSC (**b**), viscosity (**c**), and release profile (**d**) of the prepared folinic acid chitosan hydrogel.

**Figure 2 jfb-13-00142-f002:**
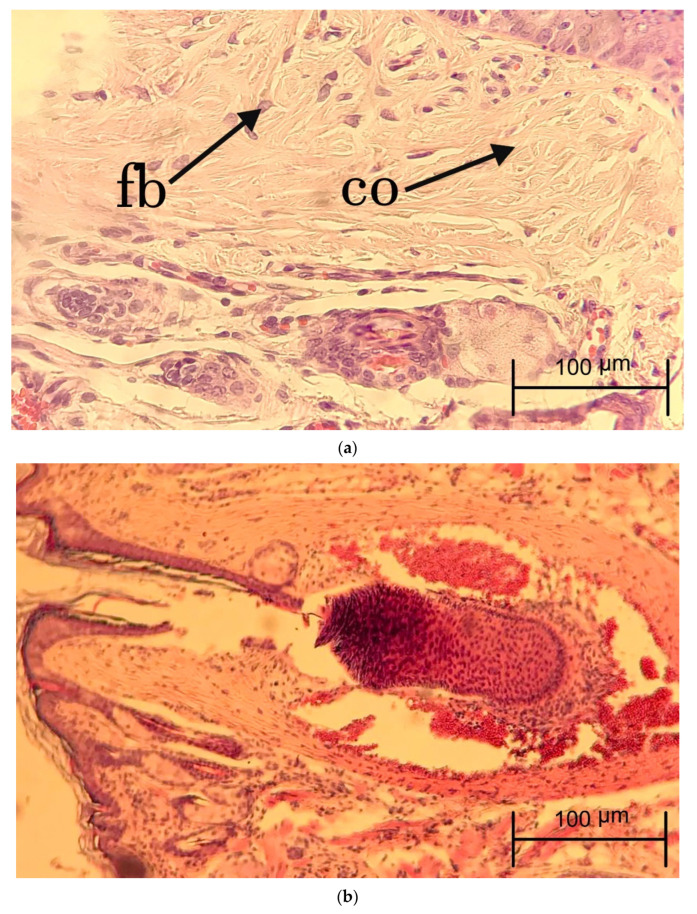
Hematoxylin and eosin staining of samples in BOT (×400 **a**), CHIT (×100 **b**), FOLCHIT (×400 **c**) groups after 14 days (fb: Fibroblast; co: Collagen fibers; bv: Blood vessels).

**Figure 3 jfb-13-00142-f003:**
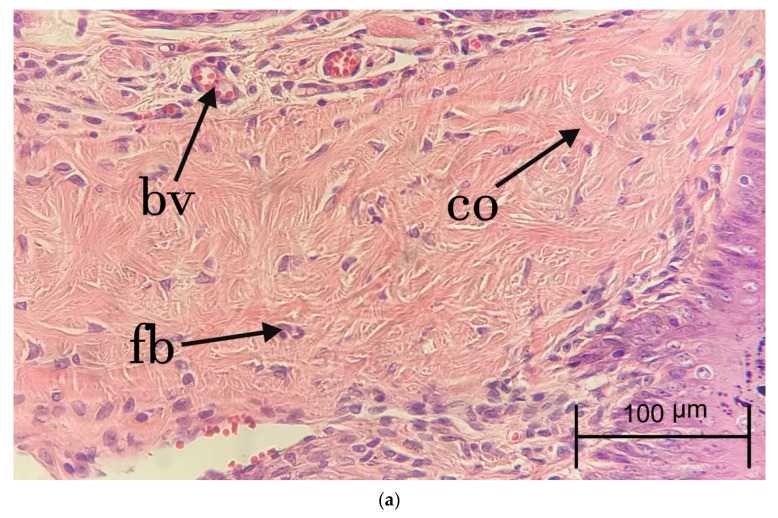
Hematoxylin and eosin staining of samples in BOT (×400 **a**), CHIT (×100 **b**), FOLCHIT (×400 **c**) groups after 28 days (fb: Fibroblast; co: Collagen fibers; bv: Blood vessels).

**Figure 4 jfb-13-00142-f004:**
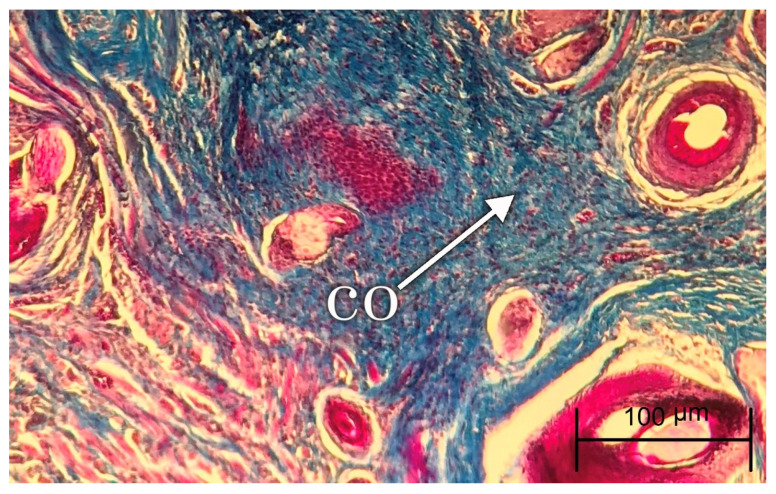
Masson’s trichrome staining of a sample in the FOLCHIT group after 28 days (co: Collagen fibers).

**Table 1 jfb-13-00142-t001:** Frequency (percentage) of scores for fibroblast proliferation in different groups at days 14 and 28.

Groups		Mild (1+)	Moderate (2+)	Marked (3+)	Mean Score	*p*-Value
CTRL	Day 14	0 (0)	5 (100)	0 (0)	2	Z = 0.00, *p* = 1.000
	Day 28	0 (0)	5 (100)	0 (0)	2	
BOT	Day 14	3 (60)	2 (40)	0 (0)	1.4	Z = 0.00, *p* = 1.000
	Day 28	3 (60)	2 (40)	0 (0)	1.4	
CHIT	Day 14	0 (0)	5 (100)	0 (0)	2	Z = 0.00, *p* = 1.000
	Day 28	3 (60)	5 (100)	0 (0)	2	
FOLCHIT	Day 14	0 (0)	0 (0)	5 (100)	3	Z = 1.96, *p* = 0.151
	Day 28	0 (0)	3 (60)	2 (40)	2.4	
Kruskal-Wallis test	Day 14	X2 = 16.48, *p* = 0.001				
	Day 28	X2 = 9.82, *p* = 0.020				

CTRL: Control; BOT: Botulinum toxin A; CHIT: Chitosan hydrogel; FOL: Folinic acid chitosan hydrogel.

**Table 2 jfb-13-00142-t002:** Frequency (percentage) of scores for collagen deposition in different groups at days 14 and 28.

Groups		Mild (1+)	Moderate (2+)	Marked (3+)	Mean Score	*p*-Value
CTRL	Day 14	0 (0)	5 (100)	0 (0)	2	Z = 0.00, *p* = 1.000
	Day 28	0 (0)	5 (100)	0 (0)	2	
BOT	Day 14	2 (40)	3 (60)	0 (0)	1.6	Z = 0.60, *p* = 0.690
	Day 28	3 (60)	2 (40)	0 (0)	1.4	
CHIT	Day 14	0 (0)	5 (100)	0 (0)	2	Z = 0.00, *p* = 1.000
	Day 28	0 (0)	5 (100)	0 (0)	2	
FOLCHIT	Day 14	0 (0)	4 (80)	1 (20)	2.2	Z = 0.65, *p* = 0.690
	Day 28	0 (0)	3 (60)	2 (40)	2.4	
Kruskal-Wallis test	Day 14	X2 = 6.18, *p* = 0.103				
	Day 28	X2 = 9.86, *p* = 0.020				

CTRL: Control; BOT: Botulinum toxin A; CHIT: Chitosan hydrogel; FOL: Folinic acid chitosan hydrogel.

**Table 3 jfb-13-00142-t003:** Frequency (percentage) of scores for infiltration of inflammatory cells in different groups at days 14 and 28.

Groups		Mild (1+)	Moderate (2+)	Marked (3+)	Mean Score	*p*-Value
CTRL	Day 14	0 (0)	5 (100)	0 (0)	2	Z = 3.00, *p* = 0.008
	Day 28	5 (100)	0 (0)	0 (0)	1	
BOT	Day 14	0 (0)	5 (100)	0 (0)	2	Z = 3.00, *p* = 0.008
	Day 28	5 (100)	0 (0)	0 (0)	1	
CHIT	Day 14	0 (0)	5 (100)	0 (0)	2	Z = 3.00, *p* = 0.008
	Day 28	5 (100)	0 (0)	0 (0)	1	
FOLCHIT	Day 14	0 (0)	3 (60)	2 (40)	2.4	Z = 2.83, *p* = 0.008
	Day 28	5 (100)	0 (0)	0 (0)	1	
Kruskal-Wallis test	Day 14	X2 = 6.33, *p* = 0.096				
	Day 28	X2 = 0.00, *p* = 1.000				

CTRL: Control; BOT: Botulinum toxin A; CHIT: Chitosan hydrogel; FOL: Folinic acid chitosan hydrogel.

**Table 4 jfb-13-00142-t004:** Frequency (percentage) of scores for vascular proliferation in different groups at days 14 and 28.

Groups		Absence (0)	Mild (1+)	Moderate (2+)	Marked (3+)	Mean Score	*p*-Value
CTRL	Day 14	0 (0)	5 (100)	0 (0)	0 (0)	1	Z = 1.50, *p* = 0.310
	Day 28	2 (40)	3 (60)	0 (0)	0 (0)	0.6	
BOT	Day 14	0 (0)	0 (0)	5 (100)	0 (0)	2	Z = 3.00, *p* = 0.008
	Day 28	0 (0)	5 (100)	0 (0)	0 (0)	1	
CHIT	Day 14	0 (0)	4 (80)	1 (20)	0 (0)	1.2	Z = 1.34, *p* = 0.421
	Day 28	1 (20)	4 (80)	0 (0)	0 (0)	0.8	
FOLCHIT	Day 14	0 (0)	0 (0)	0 (0)	5 (100)	3	Z = 3.00, *p* = 0.008
	Day 28	0 (0)	5 (100)	0 (0)	0 (0)	1	
Kruskal-Wallis test	Day 14	X2 = 17.52, *p* = 0.001					
	Day 28	X2 = 4.10, *p* = 0.251					

CTRL: Control; BOT: Botulinum toxin A; CHIT: Chitosan hydrogel; FOL: Folinic acid chitosan hydrogel.

**Table 5 jfb-13-00142-t005:** Frequency (percentage) of scores for epithelial proliferation in different groups at days 14 and 28.

Groups		Mild (1+)	Moderate (2+)	Marked (3+)	Mean Score	*p*-Value
CTRL	Day 14	5 (100)	0 (0)	0 (0)	1	Z = 3.00, *p* = 0.008
	Day 28	0 (0)	5 (100)	0 (0)	2	
BOT	Day 14	0 (0)	5 (100)	0 (0)	2	Z = 3.00, *p* = 0.008
	Day 28	0 (0)	0 (0)	5 (100)	3	
CHIT	Day 14	4 (80)	1 (20)	0 (0)	1.2	Z = 2.45, *p* = 0.032
	Day 28	0 (0)	5 (100)	0 (0)	2	
FOLCHIT	Day 14	0 (0)	0 (0)	5 (100)	3	Z = 0.00, *p* = 1.000
	Day 28	0 (0)	0 (0)	5 (100)	3	
Kruskal-Wallis test	Day 14	X2 = 17.52, *p* = 0.001				
	Day 28	X2 = 19.00, *p* < 0.001				

CTRL: Control; BOT: Botulinum toxin A; CHIT: Chitosan hydrogel; FOL: Folinic acid chitosan hydrogel.

## Data Availability

Data are contained within the article.

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
