# Peer review of "Histologic Evaluation of the Effects of Folinic Acid Chitosan Hydrogel and Botulinum Toxin A on Wound Repair of Cleft Lip Surgery in Rats"

_jfb, 2022, doi:10.3390/jfb13030142_

Round 1
Reviewer 1 Report
This manuscript contains a very interesting investigative question related to the health and social well-being of many families with newborns with cleft lip and palate. This simple and initial study opens opportunities to evaluate the nano-folinic acid in human wounds.
The experimental design was well designed and obtained significant results. Despite the few investigation tools, the experimental results were validated by the histological strategy. This reviewer suggests that the authors add molecular analyzes to evaluate the action of drugs at the metabolite and protein level for the next studies of this research group.
Few comments will be made for the revision of this manuscript.
There is no conclusion in the Abstract of this manuscript. The authors do not point out which would be the best treatment for a future clinical therapy. The authors mention the results obtained only. Thus, for researchers who are not in this area, they cannot know whether the increase in the inflammatory profile, cell proliferation, fibroblast proliferation and collagen deposition is a positive result or not.
Line 85: Replace "Acetic acid 0.1%" by "Acetic acid 0.1% (v/v)"
Line 130: What is the equipament (brand and model) that authors used to obtain the slices with thickness of five micrometers? Please, include this information.
Lines 142 to 161: All information among these lines are Results. The authors should remove this part of the Material and Methods Section and move to Results Section.
Figures 1 and 2 do not present internal scales in the images. In addition, authors should point out with asterisks some important structures indicated in the text, such as collagen deposition, fibroblast proliferation etc. Thus, Figures 1 and 2 will be more informative than they are.
Lines 253 to 266: All information contained in this lines is a resume of Results. This reviewer suggest that this paragraph will be moved to Results Section.
This reviewer considers it possible to accept this manuscript after returning the corrected version with minor corrections.
Author Response
Dear editor and reviewers,
Greetings
We would like to express our gratitude to you for your helpful and constructive comments. We really appreciate the points noted by the reviewers and think that by addressing them, our manuscript has improved significantly. In the following section, we provide a point-by-point reply to the reviewers’ comments:
Reviewer 1:
This manuscript contains a very interesting investigative question related to the health and social well-being of many families with newborns with cleft lip and palate. This simple and initial study opens opportunities to evaluate the nano-folinic acid in human wounds. The experimental design was well designed and obtained significant results. Despite the few investigation tools, the experimental results were validated by the histological strategy. This reviewer suggests that the authors add molecular analyzes to evaluate the action of drugs at the metabolite and protein level for the next studies of this research group.
Few comments will be made for the revision of this manuscript.
- There is no conclusion in the Abstract of this manuscript. The authors do not point out which would be the best treatment for a future clinical therapy. The authors mention the results obtained only. Thus, for researchers who are not in this area, they cannot know whether the increase in the inflammatory profile, cell proliferation, fibroblast proliferation and collagen deposition is a positive result or not.
Thanks for your valuable comments. We have already provided a conclusion in the abstract: “In simulated lip cleft defects, topical application of nano-folinic acid induces faster initial regeneration by higher inflammation and cellular proliferation, at the expense of a higher tendency for scar formation by slightly higher fibroblast proliferation and collagen deposition. While, injection of botulinum toxin provides less fibroblast proliferation and collagen deposition and thus lower potential for scar formation compared with the nano-folinic acid group.” However, based on this comment we added a concise conclusion at the end of the abstract (page 2, lines 46-48).
- Line 85: Replace "Acetic acid 0.1%" by "Acetic acid 0.1% (v/v)"
Done (page 3, line 88)
- Line 130: What is the equipament (brand and model) that authors used to obtain the slices with thickness of five micrometers? Please, include this information.
Leitz microtome (Oberkochen, Germany) was used for this purpose. This information is now added to the methods section (page 4, lines 132, 133).
- Lines 142 to 161: All information among these lines are Results. The authors should remove this part of the Material and Methods Section and move to Results Section.
Done (page 6, lines 179, 180)
- Figures 1 and 2 do not present internal scales in the images. In addition, authors should point out with asterisks some important structures indicated in the text, such as collagen deposition, fibroblast proliferation etc. Thus, Figures 1 and 2 will be more informative than they are.
The figures are now replaced with images of better quality with arrows indicating different elements (pages 7 and 8, figures 2 and 3). Additionally, another figure for Masson’s trichrome staining is now added as figure 4 (page 9, figure 4).
- Lines 253 to 266: All information contained in this lines is a resume of Results. This reviewer suggest that this paragraph will be moved to Results Section.
The first paragraph of the discussion is dedicated to a summary of the results, so if the respected reviewer agrees, we would like to keep it.
- This reviewer considers it possible to accept this manuscript after returning the corrected version with minor corrections
Thank you for your constructive comments!
Reviewer 2 Report
The manuscript “Histologic evaluation of the effects of nano-folinic acid chitosan hydrogel and botulinum toxin A on wound repair of cleft lip surgery in rats” was well introduced and the experimental design was adequate. The treatment comparison was based on histological analyses, however the histological examination was poorly described.
How many slides were observed for each rat ? How was examined fibroblast or epithelial proliferation ? No Ki67 labeling was indicated. If based only on cell counting, how many cells were counted and on which surface? No collagen labeling was indicated. How collagen deposition was measured (color intensity, fiber surface) ?
Figures 1a, 1b, 2a and 2b should be presented with a better quality, at least similarly to figures 1c and 2c. Moreover higher magnifications (at 400x as explained line 138) could be presented to illustrate fibroblasts, epithelial cells, inflammatory cells and vessels that were analyzed.
Legend of tables 1 to 4 should be completed. Authors should explain each abbreviation and indicate what numbers in parentheses correspond to . First line of results in table 1 indicated 5(100) meaning that the 5 rats of the control group at day 14 were classify with a moderate frequency of scores for fibroblast proliferation. However what does 100 indicate? 100% of rats with a moderate score ?
Statistical analyses can not replace a primary histological analyse with a clear identification of the counted elements.
Author Response
Dear editor and reviewers,
Greetings
We would like to express our gratitude to you for your helpful and constructive comments. We really appreciate the points noted by the reviewers and think that by addressing them, our manuscript has improved significantly. In the following section, we provide a point-by-point reply to the reviewers’ comments:
Reviewer 2:
- The manuscript “Histologic evaluation of the effects of nano-folinic acid chitosan hydrogel and botulinum toxin A on wound repair of cleft lip surgery in rats” was well introduced and the experimental design was adequate. The treatment comparison was based on histological analyses, however the histological examination was poorly described.
We have added several details for the histological examination segment (as mentioned below) based on the reviewers’ comments
- How many slides were observed for each rat?
For each rat, 1 H&E slide and 1 Masson’s trichrome slide were analyzed. This information is now mentioned in the methods section (page 4, lines 135,136).
- How was examined fibroblast or epithelial proliferation?
The approach for histological scoring in this paper was semiquantitative (based on intensity). This information is now added to the methods (page 4, lines 137, 138)
- No Ki67 labeling was indicated.
Ki67 labeling was not performed in this study.
- If based only on cell counting, how many cells were counted and on which surface?
The approach for histological scoring in this paper was semiquantitative (based on intensity) and was performed in the excised defect region. This information is now added to the methods (page 4, lines 127, 137, 138)
- No collagen labeling was indicated. How collagen deposition was measured (color intensity, fiber surface) ?
The information for Masson’s trichrome is added now in the methods section: new collagen with light blue color and mature collagen with dark blue color (page 4, lines 146-148).
- Figures 1a, 1b, 2a and 2b should be presented with a better quality, at least similarly to figures 1c and 2c. Moreover higher magnifications (at 400x as explained line 138) could be presented to illustrate fibroblasts, epithelial cells, inflammatory cells and vessels that were analyzed.
The figures are now replaced with images of better quality with arrows indicating different elements (pages 7 and 8, figures 2 and 3). Additionally, another figure for Masson’s trichrome staining is now added as figure 3 (page 9, figure 4).
- Legend of tables 1 to 4 should be completed. Authors should explain each abbreviation and indicate what numbers in parentheses correspond to.
Abbreviations are now added to the footnote of each table. Additionally, numbers in the parentheses are now explained in the caption (tables 1 to 5).
- First line of results in table 1 indicated 5(100) meaning that the 5 rats of the control group at day 14 were classify with a moderate frequency of scores for fibroblast proliferation. However what does 100 indicate? 100% of rats with a moderate score?
We have revised the tables based on the comments. Abbreviations are now added to the footnote of each table. Additionally, numbers in the parentheses are now explained in the caption to avoid confusion (tables 1 to 5).
- Statistical analyses can not replace a primary histological analyse with a clear identification of the counted elements.
We have now added several details for the histological examination and we believe that we have solved these issues.
Reviewer 3 Report
This manuscript focused on the histologic evaluation of nano-folinic acid chitosan hydrogel on wound repair of cleft lip surgery compared to botulinum toxin A. This study got some helpful results. But some details were required to be supplemented.
1. In Materials and Methods,authors showed "Preparation and characterization of nano- folinic acid chitosan hydrogel",including Fourier Transform Infrared spectroscopy (FTIR), viscosity, In vitro release profile of folinic acid.But there were no the related data.Please supplemented.
2. The characterization should be carried out to support the nano character of nano- folinic acid chitosan hydrogel.
3. Why was the 2.5% of folinic acid selected? How to determine this value?
4. It looks not in the section of Dental Biomaterials
Author Response
Dear editor and reviewers,
Greetings
We would like to express our gratitude to you for your helpful and constructive comments. We really appreciate the points noted by the reviewers and think that by addressing them, our manuscript has improved significantly. In the following section, we provide a point-by-point reply to the reviewers’ comments:
Reviewer 3:
This manuscript focused on the histologic evaluation of nano-folinic acid chitosan hydrogel on wound repair of cleft lip surgery compared to botulinum toxin A. This study got some helpful results. But some details were required to be supplemented.
- In Materials and Methods,authors showed "Preparation and characterization of nano- folinic acid chitosan hydrogel",including Fourier Transform Infrared spectroscopy (FTIR), viscosity, In vitro release profile of folinic acid.But there were no the related data.Please supplemented.
Based on this comment, supporting diagrams for FTIR, DSC, ,viscosity, and release profile are now added (page 5, lines 162,163, pages 5,6, figure 1)
- The characterization should be carried out to support the nano character of nano- folinic acid chitosan hydrogel.
The nano- prefix is removed.
- Why was the 2.5% of folinic acid selected? How to determine this value?
2.5% folinic acid was selected based on the findings of the study of Duman et al in which the authors showed that 2.5% folinic acid has higher wound healing potential in rat animal models compared with lower percentages. This reference was previously mentioned in the discussion and introduction, but now is added to the method section (page 3, lines 104, 105).
- It looks not in the section of Dental Biomaterials
Cleft lip and palate is the most common craniofacial anomaly which requires multi-disciplinary interventions by maxillofacial radiologists, orthodontists, maxillofacial surgeons, pediatric dentists, general dentists, plastic and reconstructive surgeons, speech therapists, etc. In the present study, the use of a biomaterial containing nano-folinic acid on healing of cleft lip defects in rat models was investigated which can provide useful information for the practitioners in the dental field.
Round 2
Reviewer 2 Report
Authors's answers to my comments are globally correct despite a brief description of the histological methodology used for quantififation.
Author Response
Dear editors and reviewers,
Greetings
Once again we would like to acknowledge the editors and reviewers for their constructive and helpful comments and suggestions. We think that considering these comments resulted in a better presentation of our data. Below we provided our reply to the latest comment:
Reviewer 2:
We appreciate your approval.
Reviewer 3 Report
Please describe the results showed in Fig.1
Author Response
Dear editors and reviewers,
Greetings
Once again we would like to acknowledge the editors and reviewers for their constructive and helpful comments and suggestions. We think that considering these comments resulted in a better presentation of our data. Below we provided our reply to the latest comment:
Reviewer 3:
- Please describe the results showed in Fig.1
The results in figure 1 are now explained (page 5, lines 162-176). Additionally, the relevant results are discussed in the discussion (page 16, lines 402-408)